CauseMap: fast inference of causality from complex time series

Maher M. Cyrus 1 cyrusmaher@gmail.com
Hernandez Ryan D. 2 3 4
1 Department of Epidemiology and Biostatistics, University of California , San Francisco, CA , USA
2 Department of Bioengineering and Therapeutic Sciences , USA
3 Institute for Human Genetics , USA
4 Institute for Quantitative Biosciences (QB3), University of California , San Francisco, CA , USA
Wilke Claus
Electronic publication date: 2015 Mar 5
Publication date: 2015
Volume: 3
Electronic Location ID: e824
Received 2014 Nov 3; Accepted 2015 Feb 17
Copyright: © 2015 Maher and Hernandez
Copyright year: 2015
Copyright holder: Maher and Hernandez
License: This is an open access article distributed under the terms of the Creative Commons Attribution License, which permits unrestricted use, distribution, reproduction and adaptation in any medium and for any purpose provided that it is properly attributed. For attribution, the original author(s), title, publication source (PeerJ) and either DOI or URL of the article must be cited.
License URL: https://creativecommons.org/licenses/by/4.0/

Keywords: Causality, Open source software, Time series methods, Dynamical systems, Personalized medicine

Funding: National Institutes of Health F31 Predoctoral Fellowship F31 CA180609-01 University of California, San Francisco Lloyd M. Kozloff Fellowship National Institutes of Health P60MD006902 UL1RR024131 1R21HG007233 1R21CA178706 1R01HL117004 Alfred P. Sloan Foundation Research Fellowship M. Cyrus Maher was supported by the National Institutes of Health F31 Predoctoral Fellowship (grant number 1 F31 CA180609-01), and a University of California, San Francisco Lloyd M. Kozloff Fellowship. This work was also partially supported by the National Institutes of Health (grant numbers P60MD006902, UL1RR024131, 1R21HG007233, 1R21CA178706, and 1R01HL117004) and an Alfred P. Sloan Foundation Research Fellowship to Ryan D. Hernandez. The funders had no role in study design, data collection and analysis, decision to publish, or preparation of the manuscript.

==============================
Background. Establishing health-related causal relationships is a central pursuit in biomedical research. Yet, the interdependent non-linearity of biological systems renders causal dynamics laborious and at times impractical to disentangle. This pursuit is further impeded by the dearth of time series that are sufficiently long to observe and understand recurrent patterns of flux. However, as data generation costs plummet and technologies like wearable devices democratize data collection, we anticipate a coming surge in the availability of biomedically-relevant time series data. Given the life-saving potential of these burgeoning resources, it is critical to invest in the development of open source software tools that are capable of drawing meaningful insight from vast amounts of time series data.

Results. Here we present CauseMap, the first open source implementation of convergent cross mapping (CCM), a method for establishing causality from long time series data (≳25 observations). Compared to existing time series methods, CCM has the advantage of being model-free and robust to unmeasured confounding that could otherwise induce spurious associations. CCM builds on Takens’ Theorem, a well-established result from dynamical systems theory that requires only mild assumptions. This theorem allows us to reconstruct high dimensional system dynamics using a time series of only a single variable. These reconstructions can be thought of as shadows of the true causal system. If reconstructed shadows can predict points from opposing time series, we can infer that the corresponding variables are providing views of the same causal system, and so are causally related. Unlike traditional metrics, this test can establish the directionality of causation, even in the presence of feedback loops. Furthermore, since CCM can extract causal relationships from times series of, e.g., a single individual, it may be a valuable tool to personalized medicine. We implement CCM in Julia, a high-performance programming language designed for facile technical computing. Our software package, CauseMap, is platform-independent and freely available as an official Julia package.

Conclusions. CauseMap is an efficient implementation of a state-of-the-art algorithm for detecting causality from time series data. We believe this tool will be a valuable resource for biomedical research and personalized medicine.

Introduction

Establishing health-related causal relationships is a pivotal objective in biomedical research. Yet, the interdependent non-linearity of biological systems often impedes a thorough understanding of causal dynamics. Existing and forthcoming time series data will likely play an important role in taming this complexity. Traditional cross-sectional sampling have the limitation that they may average out non-linear patterns by pooling heterogeneous signals across subjects. Long time series from a single source, on the other hand, can allow us to understand dynamic and context-specific patterns of change.

We are just beginning to grasp the biomedical relevance of such a dynamical systems perspective. Consider, for example, the human microbiome. Dysbiosis in the gut has been implicated in, e.g., irritable bowel disease (IBD), obesity, diabetes, asthma, anxiety, and depression (Foster & McVey Neufeld, 2013; Arrieta et al., 2014). Meanwhile, recent studies on microbiome dynamics have found that the ecological makeup of the human microbiome is dynamic and individual-specific. These dynamics may also interact with pathogens in interesting and therapeutically important ways. For example, there is evidence that ecological time series dynamics within the body may play a role in the progression from HIV to AIDS (Vujkovic-Cvijin et al., 2013).

Complex, dynamically evolving interdependent systems such as the microbiome pose a significant challenge to existing time series methods. Several metrics exist for detecting static non-linear relationships. These include: Spearman rank correlation, (Spearman, 1904), distance correlation (Székely & Rizzo, 2009), and mutual information content (Kullback & Leibler, 1951). Causal relationships, on the other hand, can be examined using methods such as time-lagged regression, instrumental variables, and dynamical Bayesian networks (Granger, 1969).

These causal methods are heavily model-based, however. As a result, they often falter when examining arbitrary non-linear or context-dependent relationships. Furthermore, the approaches mentioned above cannot adequately handle feedback loops, and they frequently generate both false positives and false negatives due to the influence of unmeasured confounders (Sugihara et al., 2012). These are significant liabilities, particularly in biomedicine, where relationships are usually embedded within a broad network of incompletely observed interactions.

In this paper, we present the first publicly available, open source implementation of convergent cross mapping (CCM), a model-free approach to detecting dependencies and inferring causality in complex non-linear systems (even in the presence of feedback loops and unmeasured confounding; Sugihara et al., 2012). CCM derives this power from explicitly capturing time-dependent dynamics through a technique known as state-space reconstruction (SSR). SSR has demonstrated utility for problems as diverse as wildlife management and cerebral autoregulation (Vanderweele & Arah, 2011). In practice, this analysis typically requires at least 25 data points, measured with relatively high accuracy and with sufficient density to capture system dynamics. One benefit of this approach is that, unlike most causal inference methods, the performance of CCM improves for increasingly non-linear systems. In addition, CCM can properly disentangle causal relationships that involve feedback loops, provided that strong forcing from external variables does not overwhelm the dynamics of the relationships of interest.

CCM leverages the fact that time series can be viewed as projections of higher-dimensional system dynamics (Sugihara et al., 2012). As a logical result of this property, the time series of individual variables must contain information about the full causal system. Causal dynamics (conceptualized as the state space, or manifold) can then be reconstructed using individual time series. These reconstructions can be thought of as shadows of the true causal system. If the shadows reconstructed from distinct variables can be used to predict points from each other’s time series, we can infer that these variables provide views of the same causal system and so are causally related. Since these relationships are fundamentally asymmetric, this test can also establish the directionality of causation.

Further details on CCM are available in the Supplemental Information of this paper, as well as in that of Sugihara et al. (2012). Additional explanatory resources can also be accessed through the project website (http://cyrusmaher.github.io/CauseMap.jl).

Materials and Methods

Convergent cross mapping algorithm

Consider time series of hypothetical variables X and Y. Convergent cross mapping (CCM) employs time-lagged coordinates of each of these variables to produce shadow versions of their respective source manifolds. To illustrate, suppose the time series for X were {1, 2, 3, 4}. Reconstructing a two-dimensional shadow manifold for X using a time lag of one would yield the following path: (2, 1) → (3, 2) → (4, 3). For sufficiently long time series, the path of this shadow manifold is expected to reveal important properties of the full causal system.

We will refer to the shadow manifolds reconstructed from X and Y as Mx and My, respectively. To test whether X causes Y, CCM applies the following logic: because manifold reconstruction preserves important structural components of the original system (i.e., the Lyapunov exponents; Casdagli et al., 1991), if X causes Y, then time points that are close in My should also be close in Mx. Since Mx is constructed from lags of the observations of X, the points that are close in Mx will also have similar values in the corresponding time series. Therefore, if X causes Y, then My can tell us which observations of X should best predict a given held-out point from X. Furthermore, predictability should increase with the number of manifold points in My that are considered.

Assessing predictive skill

To test whether X causes Y, My is used to infer the points in X that will best predict a given held-out point from X. We measure this performance using predictive skill, quantified by ρccm as follows. To begin, we withhold a point from X that we will then attempt to predict. We use My to infer the points in Mx that will be closest to this point of interest. This is accomplished using relative pairwise distances of corresponding points in My. We then perform a weighted average of the corresponding observations in X using exponential weights derived from these pairwise distances in My. We similarly produce predicted values for each held-out point in X. ρccm is then calculated as the Pearson correlation between held-out and predicted points. The cross validated nature of this measure serves to reduce over-fitting with respect to the model’s tuning parameters described below. To examine whether the signal converges as expected for a causal relationship, these steps are repeated using increasing numbers of points from My and Mx.

CauseMap is fast

CauseMap implements CCM in Julia, a high-performance programming language designed for facile technical computing. By way of an intelligent JIT (just in time) compilation, Julia offers much of the speed of low-level, low-productivity languages like C, while also providing the ease of use and platform independence of much slower high-level languages like Python, R, or Matlab.

At the core of CauseMap is the calculation of distances between a large number of manifold points in potentially high dimensional spaces. To optimize efficiency, CauseMap precomputes all necessary manifolds and pairwise distances using a state-of-the-art, BLAS-based protocol (for benchmarks, see: https://github.com/JuliaStats/Distance.jl).

To illustrate the speed of CauseMap as a function of time series length, in Table 1 we present the runtimes for successive catenations of the time series presented in Fig. 1. For our time series of length 71, CauseMap finishes in approximately 10 s. For a time series of over 400 observations, CauseMap still finishes in less than 20 min on a single CPU. Note that for this dataset, predictive skill was nearly perfect at a time series length of 213. This calculation finished in less than two minutes. Through this example, we observe that CauseMap can reach superb levels of performance long before increasing time series length generates significant computational challenge.

Figure 1 An example visualization from CauseMap using abundances of Paramecium aurelia and Didinium nasutum.

See Supplemental Information for more information on this system. (A) For optimal parameter values, the convergence of the cross-map correlation with library size. (B–C). The dependence of the maximum cross-map correlation on assumed dimensionality (measured by E) and the time lag of the causal effect (measured by τp). Note that the second maximum at τp = 5 corresponds to the principal frequency of the P. aurelia and D. nasutum time series, as determined by Fourier transform analysis.

Table 1 Runtime versus time series length.

Results are presented for one to six catenations of the dataset presented in Fig. 1. Runtime values are for comprehensive parameter optimizations on a single 2.6 GHz Intel Core i7 processor.

Time series length	Runtime (s)	
71	10.2	
142	40.4	
213	116.6	
284	317.2	
355	534.7	
426	1080.5	

Tuning parameter values aid causal interpretation

Beyond the speed and comparative simplicity resulting from cutting-edge JIT compilation, CauseMap offers a number of conveniences and performance enhancements. For CCM, it is particularly important to optimize two tuning parameters: E and τp.

E is the number of dimensions of the reconstructed shadow manifold. If Emax is the optimal embedding dimension, Whitney’s Theorem tells us that the dimensionality of the full causal system is generically between (Emax − 1)/2 and Emax, inclusive (Eelles & Toledo, 1992; Deyle & Sugihara, 2011). Note though that Emax is usually unknown and must be inferred from the data. This procedure is described in the following section.

τp denotes the time delay of the causal effect of interest. By examining the optimal values of these two parameters, we may place bounds on the number of variables involved in the full causal system, gain insight into the timeframe of causal effects, and obtain a built-in sensitivity analysis of the final results. The estimation of these parameters is described below.

CauseMap optimizes and visualizes tuning parameters

E and τp are optimized by multiple iterations of cyclic coordinate descent. This process chooses the values of E and τp that optimize the predictive skill of the model for held-out data points. Typically convergence of the cross map signal as a function of the time series length (L) alone is taken as the practical criterion for causality. However, measuring the dependence of this signal on E and τp is also useful for evaluating whether the result is suitably specific with respect to the assumed structure of the causal system. CauseMap therefore also includes a plotting function to visualize the dependence of the predictive skill (ρccm) on L, as well as on the joint values of E and τp.

Interpretation of output

The systematic increase of predictive skill (ρccm) with L constitutes a practical, qualitative criterion for causality (Sugihara et al., 2012). Generally, non-causal ρccm curves are flat with respect to L, while ρccm signals associated with causal signals show striking convergence given sufficient data. One exception is in the case of strong external forcing. An outside variable can introduce a cross map correlation between two quantities if it exerts a sufficiently strong influence over both. We speculate that such situations can produce ρccm values that, compared to true causal relationships, have a noisier or less interpretable dependence on E and τp. Furthermore, it is necessary to inspect the dependence of the cross map correlation on the joint distribution of E and τp in order to properly understand the meaning of the maximal values of these two variables. Note that for high throughput analyses, convergence with respect to L and sensitivity to E and τp could be assessed with, e.g., relative difference- and entropy-based measures, respectively.

CauseMap is easy to use

Beyond the tuning parameters mentioned above, CCM requires one to specify a range of library sizes, as well as the window of time points for which cross mapping should be performed. Valid values for these parameters depend in turn on E and τp. To reduce complexity for the user, CauseMap calculates intelligent defaults for these parameters, while also offering the option of specifying them directly.

Caveats and considerations

The strengths and weaknesses of CCM make it nicely complementary to the existing tools for causal inference. Unlike most algorithms for this task, the performance of CCM improves for increasingly non-linear systems. However, this capacity depends upon relatively long time series. CCM requires at least 25 data points, measured with relatively high accuracy and with sufficient density to capture system dynamics.

There are also theoretical and practical limitations to the types of relationships that CCM can disentangle. For example, if both X and Y are almost entirely determined by a third variable Z, we would be at risk of inferring a spurious relationship between X and Y (as we would be with any other causal inference method). If the forcing from Z is relatively weak, however, CCM is expected to provide a lower false positive rate relative to other methods (Sugihara et al., 2012).

CCM also examines relationships between variables in a pairwise fashion. However, by leveraging dynamical systems theory, it has the ability to measure possibly bidirectional causal effects even in the presence of unmeasured confounding. Finally, CCM performs best with complete data sampled at regular intervals. This is particularly important for inferring the time lag of the causal effect. This limitation can be partially addressed through filtering or appropriate interpolation of input data.

Results and Discussion

To demonstrate CauseMap’s functionality and performance, we examined the predator–prey relationship between Paramecium aurelia and Didinium nasutum (Heskamp et al., 2013). Observations were collected every 12 h for 30 days, yielding a total of 60 data points (Veilleux, 1976). Plotted in Fig. 1 is the CauseMap visualization of the dependence of predictive skill (ρccm) on L, E, and τp. In Fig. 1A, we observe convergence in ρccm with respect to L, the number of data points used for prediction of held-out observations. This convergence is a practical criterion for causality and the source of the name convergent cross mapping.

The interpretation of this result is that the causal relationship between P. aurelia and D. nasutum is bi-directional. That is, the number of predators influences the number of prey, and vice-versa. Furthermore, relative strengths of convergence indicate that the top-down influence of the predator (D. nasutum) is stronger than the bottom-up influence of the prey (P. Aurelia). As pointed out by Sugihara et al., this finding is consistent with experimental results and illustrates the ability of CCM to investigate asymmetrical bi-directional coupling in non-linear systems.

Figures 1B and 1C show the dependence of the max ρccm on E (the dimensionality of the reconstructed system), and the supposed time lag of the causal effect (τp). Overall, the patterning of these heatmaps demonstrates that max ρccm has a reasonable and moderately specific dependence on the dimensionality of the reconstructed system (E) and on the time lag of the causal effect (τp). We expect this built-in sensitivity analysis to rule out some cases of spurious convergent signal caused by external forcing. In addition, this analysis can alert the researcher when alternative combinations of E and τp explain the data approximately as well as the optimal values of E and τp.

For the system presented in Fig. 1, while the max ρccm is relatively insensitive to the assumed dimensionality, the best-performing τp values correspond to either immediate causal effects, or those delayed by five days. Note that τp = 5 corresponds to the principal frequency of the Paramecium aurelia and Didinium nasutum time series, as determined by Fourier transform analysis (see Supplemental Information for further details). This suggests that the peak at τp = 5 is artifactual. Therefore, we are able to infer from the data that, as we would expect, predator and prey populations exert bidirectional effects in real-time.

Performance

Approximately 100 CCM evaluations were conducted to produce Fig. 1. These calculations finished in approximately 10 s on a single 2.6 GHz processor. Each of these evaluations involved the prediction of over 60,000 points, compiled across all sliding windows of libraries of varying lengths. At an average of 1.7 ms per prediction, this is a highly efficient implementation given the computational challenges.

Dependence of predictive skill on time series length

CauseMap is designed to examine causal relationships in time series with 25 or more observations. In order to illustrate the effects of shorter time series, we thinned the Paramedium-Didinium data set by one-half and by one-third, yielding series of 30 and 20 observations, respectively. Figure 2 demonstrates the effect of this reduction on the convergence of predictive skill (ρccm). We see that the 1/2 thinned data set recapitulates the trends observed in the full series, including the relative magnitudes of ρccm between the mappings of Didinium to Paramecium and vice versa. The 1/3 thinned sample set, on the other hand, no longer demonstrates convergence. In addition, compared to the longer sets, it exhibits the opposite trend in relative predictive skill between the two mappings. Patterns in max ρccm versus E and τp are approximately conserved, however (Fig. S1).

Figure 2 The effect of time series length on ρccm convergence.

Black, blue, and red lines illustrate ρccm for the full, 1/2 thinned, and 1/3 thinned datasets, respectively. For a given color, darker lines show ρccm for the test of whether Didinium abundance influences Paramecium abundance. Lighter lines examine the converse.

This example illustrates that CCM performance drops off sharply between 20 and 30 data points. This behavior is partially due to the fact that the predictive skill for a given library size is averaged across sliding windows of that size. As time series get shorter, there are fewer windows of appropriate size across which to average, so the estimate for predictive skill becomes much less reliable.

Potential Biomedical Applications

Despite its requirement for relatively long time series (>25 observations), CauseMap has the advantage of requiring only a single time series for each variable. In dynamical systems with widely varying or context-specific behavior, this would allow researchers to draw conclusions that are tailored to a given patient, for example. Rather than acting on population averages, biomedical researchers would be free to fully personalize therapy to the unique biology and ecology of the patient. One example of this is in the treatment of microbiome dysbiosis. Imbalances in the microbiome have been implicated in, e.g., irritable bowel disease (IBD), obesity, diabetes, asthma, anxiety, and depression. While fecal transplantation therapy is effective in treating specific types of dysbiosis, next generation therapeutics may offer a blend of purified strains, tailored to the gut ecology of the patient. We believe CauseMap has the potential to be a valuable tool for designing such breakthrough therapies.

Additional examples include understanding patient-to-patient variability in drug response using time series metabolomics, and examining the basis of, for example, influenza seasonality using global time series. We expect that such applications will continue to proliferate as the costs of data collection decrease over the coming years. For this reason, we believe it is vitally important that the biomedical research community have access to an efficient implementation of CCM that is user-friendly and available for immediate field testing.

Planned future development

In future versions, we will include S-map calculations to evaluate the non-linearity of the causal system. We will also add a bootstrap-based procedure for library selection, as opposed to the current approach using sliding windows. This has been shown to reduce the effect of secular trends on the cross map correlation (H Ye & G Sugihara, pers. comm., 2014). In addition, we will re-implement the plotting functionality in Julia, removing the requirements of Python and matplotlib for visualization. Finally, we will design Python and R wrappers for CauseMap functions so that our codebase can be easily leveraged from those environments as well. User suggestions will also be considered as we decide how best to develop the tool.

Conclusions

CauseMap provides a fast, user-friendly implementation of CCM, a powerful new method for exploring dependencies and even establishing causality in complex, highly non-linear datasets with many unobserved variables. We believe that CCM holds a great deal of promise for a wide range of applications, including personalized microbiome therapy and metabolic dynamics analysis. As novel time series datasets continue to emerge, it is our hope that CauseMap will allow researchers to uncover interesting and biomedically actionable causal relationships using this next-generation time series method.

Supplemental Information

Supplemental Information Supplemental materials

Click here for additional data file.

Figure S1 The maximal predictive skill as a function of E, τp, and the number of included points

Click here for additional data file.

We would like to thank George Sugihara, Hao Ye, and Ethan Deyle for their invaluable help in understanding the core details of the CCM algorithm, and Lawrence Uricchio, Nicolas Strauli, and Raul Torres for comments on this manuscript.

List of abbreviations

CCM Convergent cross mapping

SSR State space reconstruction

Additional Information and Declarations

Competing Interests

Author Contributions

Data Deposition

The authors declare there are no competing interests.

M. Cyrus Maher conceived and designed the experiments, performed the experiments, analyzed the data, contributed reagents/materials/analysis tools, wrote the paper, prepared figures and/or tables, reviewed drafts of the paper.

Ryan D. Hernandez contributed reagents/materials/analysis tools, wrote the paper, prepared figures and/or tables, reviewed drafts of the paper.

The following information was supplied regarding the deposition of related data:

All examples and associated data are available in the project repository (https://github.com/cyrusmaher/CauseMap.jl).

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
