# Peer review of "CauseMap: fast inference of causality from complex time series"

_PeerJ, doi:10.7717/peerj.824_

## Round 0.1 · original submission · Major Revisions

Both reviewers see value in your CauseMap program but feel the manuscript is unclear and omits important information. By my reading of your manuscript, the reviewers are completely correct. Therefore, I'd like to ask you to prepare a revised version of the manuscript that addresses all the reviewer comments.

·

Basic reporting

This paper presents an open-source implementation of a convergent
cross mapping (CCM) - a previously-published method for causal
analysis of time series data. The authors discuss an implementation of
this algorithm as a Julia package, along with a presentation of some
of the key parameters and how they might best be adjusted. An example
of the impact of these parameters, and of the length of the time
series, on the analysis of the relationship between two
micro-organisms is presented.

This topic seems to be of interesting, and the CauseMap
implementation might prove very useful. Unfortunately, the
presentation of this work leaves much to be desired. There was no
discussion of the underlying algorithm -readers are simply referred to
the original paper for details. The description of the parameters was
particularly opaque, as we are told only that "E is related to the
dimensionality of the full causal system". The discussion of the
predictive skill does not define the method that is used, and the
example provided in Figure 1 does not demonstrate the inference of
causality that is presumably the point of the example provided.

I have no doubt that the implementation might be of real value, but a
clearer presentation is needed to make this paper more compelling.

Experimental design

The experimental design seems to be based on the evaluation of the
values of the parameters and of the impact of the length of the time
series. These analyses seem useful in terms of tuning the algorithm,
but there is nothing in the way of validation of the causality
algorithm itself.

Validity of the findings

Although the discussion of the values of the parameters and of the
effect of the time series seems plausible enough, the conclusions that
might be draw from the example in Figure 1 are unclear, leaving the
validity in question.

·

Basic reporting

I'm evaluating this manuscript with the assumption that it's purpose is to communicate the value of this software to potential users.

First, I think more introduction and background is needed for readers to be able to tell if CauseMap is really the right tool for their job. The original article on convergent cross mapping by Sugihara and coauthors is more cautious about the applicability of the method, and contains guidance that bears repeating here such as: "In general, state space reconstruction (SSR) methods work best when the system is nonlinear and can be approximated in few dimensions, and especially when observational noise is not excessive." Another limitation that the authors might mention is that detection of directional causal relationships is not possible when very strong forcing induces synchrony.

Second, I think the readers may want to know more about the acceptable input for this program. I suspect many users may have data with missing observations and observations that occur at varying intervals. Perhaps the authors could comment on whether the software could handle this or how much work extension to this case might be to user/developers. Also, although the minimum length of the time series is demonstrated, there is no clear indication of how much data this method can handle. One might like to see a plot of run time against time series length or the dimension of the shadow manifold, for example.

Finally, more guidance on interpretation of the output seems advisable. I think most readers would appreciate a more detailed walk-though that describes what to look for in the output. For example, it would be nice if the authors could suggest a specific criterion of acceptable convergence, and what might be a sign in the output that something is wrong with the analysis. How should users interpret the converged statistics measuring predictive skill?

It seems appropriate to cite B. G. Veilleux, thesis, University of Alberta (1976) as the source of the experimental data at some point.

I quote from the supplement: "Because manifold reconstruction preserves the Lyapunov exponents of the original system [18]." But there is no bibliography in the supplement that I can look up reference 18 in, nor are the references in the main text's bibliography numbered.

The figure labeling has some problems. In Fig. 1A there is a typo in the y-axis label. tau_s appears in the legend of that same figure but is not defined in the text or caption. The positioning of the letters that indicate subfigures in the upper right rather than the upper left is a bit confusing. I would suggest a descriptive label accompany the symbols used on all axis labels wherever space permits.

Experimental design

I was not able to reproduce Figure 1 using the quick start guide on the project's website. I had to look in the repository to find out how to load the data and then I got an error about there being no matching method for makeoptimizationplots(). For a publication like this, it seems especially desirable for readers to do such a thing and for such an example to be included as supplementary information with instructions on running it.

Validity of the findings

The authors look at the sensitivity of predictive skill to E and tau_p and report the optimization over these parameters and plotting of predictive skill with respect to these parameters as a feature of the program. However, there is no reference given to explain why these features are important. I don't find the example in Figure S1 data to be very convincing of the statement on line 138 that this provides a more robust measure of causality. There is not really any strong dependence on E show in any of the figures. The fact that tau_p=5 leads to high predictive skill seems to me be explained by it being the typical period length of the population cycles, which can be seen simply by looking at a plot of the time series. I think the claim about using variation over E and tau_p to judge causality need to be more clearly backed up or identified as speculation.

Additional comments

Overall, CauseMap looks like a nice piece of software and I hope my comments help you in communicating it's value.

---

## Round 0.2 · Minor Revisions

Both reviewers request additional minor revisions. I broadly agree with their assessment. In particular, as reviewer 1 states, writing this paper as clearly as possible is fundamentally to your own benefit. The better you can delineate when exactly this method is useful and when it is not, and how exactly it is used, the more you can expect other people to build on your work.

·

Basic reporting

Although this version of the paper is much improved, I feel that this revision lags in terms of clarity and detail - as Reviewer 2 said, more detail would be needed to understand the utility and applicability of the tool.

Specifically, the expanded description of the E parameter, no guidance is given regarding the value of Emax. How should this parameter be chosen?

I am also curious about the lengths of relevant time series - this paper talks in terms of 25 time points as "relatively long" , but many time series (for example, EKG) are much longer - aside from expected scaling issues, does the algorithm still work?

Experimental design

The evaluation seems reasonable, if a bit preliminary. I was frankly confused by much of the discussion of the parameter values. I was also left unclear as to the limitations of the algorithm - can CauseMap be applied to time series with multiple potential causal relationships, or is it simply a comparison of two inputs?

Validity of the findings

I see no particular questions regarding the validity.

Additional comments

Given that the purpose of this paper is to present an implementation of a potentially useful algorithm, I certainly do not expect the authors to provide either a full description of all of the details, or a defense of all of the shortcomings. However, the paper as presented does not describe the parametrization clearly, and the limits and the strengths of the algorithm are not clearly presented. Clarification of these issues would greatly increase the value of this paper.

·

Basic reporting

Three statements in the introduction require references:

1. Page 2, line 32, "For example, there is evidence that ecological times
series dynamics ... AIDS."

2. Page 3, line 40, "As a result, the often falter when ... relationships."

3. Page 3, line 41, "Furthermore, the approaches mentioned ... confounders.

Please capitalize "spearman" and "fourier" when it is standard practice to do so.

Experimental design

No Comments

Validity of the findings

I'm concerned about the technical accuracy of the sentence on page 7: "If E_max is the optimal embedding dimension, Whitney's Theorem tells us that the dimensionality of the full causal system must be between (E_max - 1)/2 and E_max inclusive." According to page 3 of Deyle and Sugihara (Generalized Embedding Theorems, PLoS One, 2011) paper , Whitney's theorem does not by itself say that embeddings will exist in the set of time-delay embeddings, which are what this procedure considers. Also, it seems to me that this statement presumes that E_max is always an accurate estimate of the true embedding dimension, while I would expect that accuracy would be data- and system-dependent. Please clarify.

Additional comments

In general, I am satisfied with the authors' revisions. However, I still see
the need for a few more changes.

---

## Round 0.3 · accepted · Accept

Thank you for addressing the remaining reviewer comments.